# Real-Time Impedance-Based Monitoring of the Growth and Inhibition of Osteomyelitis Biofilm Pathogen *Staphylococcus aureus* Treated with Novel Bisphosphonate-Fluoroquinolone Antimicrobial Conjugates

**DOI:** 10.3390/ijms24031985

**Published:** 2023-01-19

**Authors:** Parish P. Sedghizadeh, Philip Cherian, Sahar Roshandel, Natalia Tjokro, Casey Chen, Adam F. Junka, Eric Hu, Jeffrey Neighbors, Jacek Pawlak, R. Graham G. Russell, Charles E. McKenna, Frank H. Ebetino, Shuting Sun, Esmat Sodagar

**Affiliations:** 1Division of Periodontology, Diagnostic Sciences and Dental Hygiene, Infection and Immunity Laboratory, Herman Ostrow School of Dentistry, University of Southern California, Los Angeles, CA 90007, USA; 2BioVinc LLC, Pasadena, CA 91107, USA; 3Department of Pharmaceutical Microbiology and Parasitology, Medical University of Wroclaw, 50-367 Wroclaw, Poland; 4Department of Pharmacology and Medicine, Pennsylvania State University College of Medicine, Hershey, PA 17033, USA; 5Medical Department, Lazarski University, 02-662 Warsaw, Poland; 6The Botnar Research Centre, Nuffield Department of Orthopaedics, Rheumatology and Musculoskeletal Sciences, University of Oxford, Oxford OX1 2JD, UK; 7The Mellanby Centre for Bone Research, Department of Oncology and Metabolism, University of Sheffield, Sheffield S10 2TN, UK; 8Department of Chemistry, University of Southern California, Los Angeles, CA 90007, USA

**Keywords:** bisphosphonate, antibiotic, fluoroquinolone, conjugate, biofilm, *Staphylococcus*, impedance, bone targeting

## Abstract

Osteomyelitis is a limb- and life-threatening orthopedic infection predominantly caused by *Staphylococcus aureus* biofilms. Bone infections are extremely challenging to treat clinically. Therefore, we have been designing, synthesizing, and testing novel antibiotic conjugates to target bone infections. This class of conjugates comprises bone-binding bisphosphonates as biochemical vectors for the delivery of antibiotic agents to bone minerals (hydroxyapatite). In the present study, we utilized a real-time impedance-based assay to study the growth of *Staphylococcus aureus* biofilms over time and to test the antimicrobial efficacy of our novel conjugates on the inhibition of biofilm growth in the presence and absence of hydroxyapatite. We tested early and newer generation quinolone antibiotics (ciprofloxacin, moxifloxacin, sitafloxacin, and nemonoxacin) and several bisphosphonate-conjugated versions of these antibiotics (bisphosphonate-carbamate-sitafloxacin (BCS), bisphosphonate-carbamate-nemonoxacin (BCN), etidronate-carbamate-ciprofloxacin (ECC), and etidronate-carbamate-moxifloxacin (ECX)) and found that they were able to inhibit *Staphylococcus aureus* biofilms in a dose-dependent manner. Among the conjugates, the greatest antimicrobial efficacy was observed for BCN with an MIC of 1.48 µg/mL. The conjugates demonstrated varying antimicrobial activity depending on the specific antibiotic used for conjugation, the type of bisphosphonate moiety, the chemical conjugation scheme, and the presence or absence of hydroxyapatite. The conjugates designed and tested in this study retained the bone-binding properties of the parent bisphosphonate moiety as confirmed using high-performance liquid chromatography. They also retained the antimicrobial activity of the parent antibiotic in the presence or absence of hydroxyapatite, albeit at lower levels due to the nature of their chemical modification. These findings will aid in the optimization and testing of this novel class of drugs for future applications to pharmacotherapy in osteomyelitis.

## 1. Introduction

A biofilm is a complex microbial community, composed of one or more bacterial species attached to a solid substrate and embedded in a matrix. Biofilms are more resistant to antibiotics, biocides, and host defense strategies [1,2]. Biofilm pathogens play a critical role in the pathogenesis of many human diseases including chronic infections of hard and soft tissues, certain cancers, and even conditions such as Alzheimer’s disease to name a few [3,4].

*Staphylococcus aureus* is a predominant cause of biofilm-associated infections, including more serious infections to the host tissues such as endocarditis and osteomyelitis [5]. Osteomyelitis is a limb- and life-threatening orthopedic infection and is extremely challenging to treat clinically [6]. We recently demonstrated that osteomyelitis pathogens such as *S. aureus* can invade bone, establish chronic biofilms throughout three-dimensional bony surfaces, and directly destroy and resorb bone without the participation of host immunity or osteoclastogenesis [7]. The inadequate efficacy of current antimicrobial treatments for osteomyelitis has been ascribed to the limited access of systemically administered antibiotics to osseous sites where causative bacteria reside as biofilms, including surfaces deep within the osteocytic canalicular network [8]. Still the most efficient treatment for osteomyelitis is intravenous antibiotic therapy and surgical debridement of the infected area [9]. However, surgery is not possible in some patients and, when it is possible, it can be associated with significant complications. Furthermore, antibiotics at high intravenous doses for extended periods of time are often needed clinically because antibiotics in general have poor bone pharmacokinetics and causative biofilm pathogens have higher resistance than their planktonic counterparts [10,11]. Systemic antibiotic therapy can also be associated with adverse effects in some patients. Therefore, it is necessary to explore new therapeutic antibiotics to improve bone bioavailability and safety in osteomyelitis treatment or pharmacotherapy.

To this end, our team has been designing and testing novel bone-targeting antibiotic drugs utilizing the conjugation of antibiotics to bisphosphonate (BP) compounds via the designed releasable linkers [12]. The BPs are a class of compounds that have been known as bone-targeting moieties. BPs bind to the hydroxyapatite (HA) matrix, the main inorganic component in hard tissues such as bone, via strong dentate ionic bonds. Some BPs have been developed and extensively utilized clinically to treat resorptive or degenerative bone diseases including osteoporosis, multiple myeloma, metastatic cancer to bone, and Paget’s disease [13,14]. Due to this specific and strong bone binding affinity of BPs, we designed and tested the novel BP-quinolone antibiotic conjugates with bone targeting property for activity against *Staphylococcus aureus* biofilms.

Furthermore, besides the lack of effective antibiotics for the treatment of biofilm-related diseases, e.g., osteomyelitis, as discussed above, new technologies to facilitate the study of real-time bacterial growth and development is also of great importance to help in understanding pathogenesis as well as to screen new therapeutic drugs. Currently, microscopy techniques have been extensively used for the quantification of detached bacteria and biofilm observations, serving as biofilm progression monitoring methods; however, these are limited by high labor intensity, intrusive sampling, and/or long-time lags from sampling to obtaining a result [15]. Thus, there is an urgent need to develop new monitoring assays that are sensitive, reproducible, and precise for biofilm-associated research and biomedical applications. In the present study, we assessed real-time electrode impedance measurements to test the antimicrobial efficacy and effects of our novel antibiotic conjugates on *Staphylococcal aureus* growth and formation in comparison to the parent or non-conjugated antibiotic compounds as controls. Gold microelectrodes embedded in the bottom of microtiter wells can non-invasively monitor bacterial statuses including cell number or index, shape and size, and attachment—all without the need for labeling. The adhesion of bacteria to gold microelectrodes impedes the flow of currents, allowing for the assessment and quantification of microbial growth and death. Impedance assay measurements can detect whether these bacteria are unaffected, inhibited, or induced during antimicrobial therapy in real time as shown herein.

Impedance technology detects changes in the flow of current between the electrodes that are located on the bottom of a well. When applied to the context of biofilm-producing bacteria, the Cell Index growth curves display an exponential shape that are similar to other conventional end-point assays such as the classical microtiter adherence end-point assays with safranin staining. Furthermore, the Cell Index values increase when bacteria are grown in the presence of high sugar in the medium, while the sterile medium has no effect on the Cell Index values through time. When proteinic biofilm strains are treated with proteinaseK, a marked decrease in the curve slope is observed, suggesting a direct correlation between the extent of the growth and Cell Index values. Because of the positive effect of the cell number and extracellular matrix content on impedance, the Cell Index value represents a measurement of total bacterial or biofilm mass and this is well established in the literature [15,16], but it cannot discriminate between the components associated with the cell number (cell growth) and the components due to the extracellular matrix (extracellular biofilm production), so it will not detect effects that are specific for either component. Additionally, because of the small contribution of bacterial biofilms to the impedance value and relatively smaller variations in the Cell Index, when compared to adherent mammalian cells, the correct equilibration of the medium before performing a background reading and proper negative controls are very important [16].

## 2. Results

### 2.1. Cell Suspension Concentration Optimization

A serial dilution of cell suspension was tested prior to antibiotic testing to determine the most optimal culture conditions. The optimal conditions will allow sufficient time for the cells to grow to saturation from the early lag phase. From our results, shown in Figure 1, it was determined that a 1:200 dilution of the cellular suspension was optimal so that we could see the progression of cell growth over the testing period.

### 2.2. Evaluating the Effectiveness of Testing Compounds on Biofilm Growth Inhibition

#### 2.2.1. Preventative Effect of Testing Compounds on Biofilm Growth in the Absence of HA

To study the antibacterial activity to prevent the biofilm formation of *S. aureus*, real-time dose–response experiments were conducted using BP-antibiotic conjugates and non-conjugated antibiotics against *S. aureus*. To study their antibacterial activity to prevent biofilm formation, the testing compounds were added to the bacterial inoculum from the beginning of the experiment. The preventative effect of different compounds on the tested strain is summarized in Figure 2. As shown in Figure 2a–d, the lowest concentration of testing compounds (2 µg/mL of ECC and ECX, 0.1 µg/mL of C, 0.025 µg/mL of X) was not able to alter the biofilm growth. At higher concentrations (5 µg/mL of ECC and ECX, 0.25 µg/mL of C, 0.2 µg/mL of X), there was at least a 50% reduction in the biofilm formation relative to the control and it seemed that this concentration provided a temporary suppression of the biofilm formation. At 10 µg/mL of both ECC and ECX, 2 µg/mL of C, and 0.5 µg/mL of X, the bacterial biofilm growth stopped altogether. This was also evident for BCN and BCS as shown in Figure 2e,g as concentrations lower than 3 µg/mL did not inhibit biofilm growth. For sitafloxacin and nemonoxacin, a concentration of 0.02 µg/mL completely inhibited biofilm growth.

#### 2.2.2. Preventative Effect of Testing Compounds on Biofilm Growth in the Presence of HA

In another set of preventative experiments, the BP-antibiotic conjugates and parent antibiotics were premixed with HA and then the mixture was added to the bacterial inoculum from the beginning of the experiment. Figure 3 shows that their inhibitory efficacy on biofilm growth is concentration dependent; the lowest concentration of the conjugates and parent antibiotics used in our assay was not enough to impact biofilm growth. Increasing the testing concentrations resulted in the reduction in biofilm growth; the biofilm growth was completely inhibited at the following concentrations: 10 µg/mL of ECC, 15 µg/mL of ECX, 10 µg/mL of BCN, 5 µg/mL of BCS, 0.25 µg/mL of C, 0.1 µg/mL of X, 0.08 µg/mL of N, and 0.05 µg/mL of S.

### 2.3. Calculation of the MIC50

To compare the efficacy of the BP-antibiotic conjugates to the parent antibiotics, a numerical parameter from the baseline delta cell index versus the log of concentration curves was established after defining the best time point for each testing compound. The MIC50 of tested antibiotics and BP-antibiotic conjugates against *S. aureus* in prevention assays in the presence and absence of HA is presented in Table 1.

### 2.4. Bio-Screen Study of Biofilm Growth

In order to confirm the results from the xCELLigence RTCA and to assess whether the BP and etidronate moieties themselves have any antimicrobial activity against *S. aureus*, a set of experiments was performed using the Bio-Screen C automated microbiology growth curve analysis system. In these experiments, different concentrations of each bisphosphonate were tested in comparison to the BP-antibiotic conjugates and parent antibiotics (BCN, BCS, ECC, ECX, C, N, S, and X). Bio-Screen C analysis monitors bacterial growth and/or survival by following changes in the culture turbidity as measured by the changes in the optical density (OD). Higher bacterial growth and survival correlate with higher OD readings. As shown in Figure 4a–c, etidronate and BP at 5 and 10 µg/mL did not have significant effects on bacterial growth, thus any changes in bacterial growth in the presence of BP-antibiotic conjugates should not be due to BP and etidronate moieties. The Bio-Screen C results showed that the following antibiotics suppressed the growth of *S. aureus*: BCS (2 µg/mL), BCN (5 µg/mL), Nem (0.03 µg/mL), Sita (0.01 µg/mL), ECX (2 µg/mL), M (0.5 µg/mL), and Cipro (0.05, 0.1, 0.2 µg/mL). These antibiotics did not kill *S. aureus*, but it either required a significantly longer period to reach the same OD as the untreated group or it reached a significantly lower OD compared to the same control group. On the other hand, these antibiotics were found to kill *S. aureus*: Sita (0.03 µg/mL), BCS (6 µg/mL), Nem (0.06 µg/mL), ECX (5 µg/mL), M (0.75 µg/mL), ECC (10 µg/mL), and Cipro (0.5 µg/mL), while the rest of the antibiotics tested did not have any effects on the growth and survival of *S. aureus*. These observations were similar to the ones from our xCELLigence experiments; thus, the result confirmed our findings from real-time analysis and validated the accuracy of the real-time analysis.

### 2.5. Affinity of Tested Compounds to HA

With the antimicrobial efficacy of the BP-antibiotic conjugates established, we aimed to evaluate the affinity of the tested antibiotics and conjugates to HA using HPLC methodology. The obtained results were expressed as [%] affinity and were as follows: BCS = 93% > BCN = 81% > X = 40% > S, N, C ≤ 37%. The highest affinity to HA was seen with BCS and BCN as compared to the parent antibiotics alone or the etidronate conjugates.

### 2.6. Preventative Antimicrobial Assays through Spectroscopic Analysis

Next, a series of antimicrobial tests for all the BP-antibiotic conjugates were performed in a preventative experimental setting using the traditional spectroscopic analysis [12]. The HA spherules were added to various concentrations of the BP-antibiotic conjugates and then inoculated with *S. aureus* for 24 h; quantitative assessments indicated a drastic reduction in bacterial growth from 50 to 200 μg/mL for BCS, from 100 to 200 μg/mL for ECX, and at 200 μg/mL for BCN and ECC, as shown in Figure 5.

## 3. Discussion

The ultimate goal of drug design is to identify therapies that work directly at the tissue, cell, and biochemical target of any specific disease but do not affect the biochemistry at any other non-diseased compartment of the body. Calcified tissues of the skeleton are targeted with great specificity by the BP drug class to address bone-related diseases with minimal soft tissue drug side-effects. In previous work, we demonstrated the in vivo efficacy of BP-conjugated compounds for the treatment of skeletal conditions such as osteomyelitis and multiple myeloma [11,17]. Wang et al. linked a clinical drug used in myeloma treatment (Bortezomib) to a BP that binds avidly to bone, but is not anti-resorptive, using a novel linker to generate a BP-linked Btz (named BP-Btz) conjugate and demonstrated that BP-Btz, but not Btz alone, bound to bone slices and inhibited the growth of myeloma cells in vitro. In vivo, BP-Btz more effectively reduced tumor burden and bone loss than Btz in the 5TGM1 mouse model of multiple myeloma; additionally, BP-Btz generated significantly less in terms of systemic adverse effects, such as thrombocytopenia, compared with Btz alone [18]. In the osteomyelitis context, we have demonstrated that bone-targeted systemic treatment using a BP-ciprofloxacin conjugate (BCC) resulted in greater efficacy in vivo than the parent antibiotic alone [12]. The quantitative determination of the colony forming units (cfu) of bacteria from resected tissue from a peri-prosthetic osteomyelitis model showed that a single dose of 10 mg/kg of BCC provided circa 6 log units of greater suppression of bacteria relative to an untreated control, while multiple doses of the parent antibiotic ciprofloxacin (6 times higher total molar dose compared to BCC) showed 2–3 (from 99% to 99.9%) log units of bacterial suppression [12].

The purpose of the current work was to study, for the first time, the effects of BP-conjugated antibiotics on the dynamics of bacterial growth and biofilm inhibition with real-time impedance monitoring using the model osteomyelitis pathogen *S. aureus*. We utilized impedance measurements in microtiter plates with gold electrodes to assess these effects under different experimental conditions. This high-throughput technology allows for the real-time monitoring of bacterial growth, enabling the study of the dynamics of biofilm formation and eradication. The results obtained in this work and by others support that impedance-based real-time monitoring is fast and reliable while producing reproducible and consistent results and can be effectively used to perform screening of bacteria that are able to form biofilms [16]. Further, it allows for the testing of compounds that interfere with bacteria growth and biofilm formation, or remove attached pathogens, upon abiotic (gold) surfaces; this methodology also enables the determination of effective antimicrobial concentrations [19]. Additional advantages of the impedance-based assay, compared to the standard methods applied to biofilm research, are that it is not an end-point method because it continuously monitors growth and it is a label-free technique given that impedance curves are denoted in real-time without the need for staining procedures [16]. Despite all the aforementioned advantages, there are some limitations in using impedance-based measurements; for example, in order to measure the total biofilm biomass, the technology cannot discriminate between the components associated with cell number (cell growth) and those due to the extracellular matrix (extracellular biofilm production), so it will not detect effects that are specific for one component. Additionally, because of the small contribution of bacteria biofilms to the impedance value and relative smaller variations in the Cell Index, when compared to adherent mammalian cells, the correct equilibration of the medium before performing the background reading and proper negative controls are very important.

Herein, we tested newer generation and potent quinolone antibiotics and used these for conjugation to the BP moieties. This bisphosphonate-conjugated antibiotics have shown activity against S. *aureus* pathogens previously via in vivo target and release studies [20]. Quinolones are also used in osteomyelitis treatment clinically for their effectiveness against causative pathogens [21]. The collected results are mainly indicative of activity against planktonic bacterial cells and not necessarily against bacterial biofilms, although similar data have been reported as biofilm specific in the previous literature. Biofilm analysis requires different experimental conditions (e.g., inoculum higher than 10^7^ CFU/mL; staining procedures with for example crystal violet or SYTO-9; and antibiotic concentration lower than MIC100) in order to be detected. Our results show that the parent antibiotics and the BP-antibiotic conjugates were effective in inhibiting bacterial growth and formation. Sitafloxacin and nemonoxacin demonstrated the lowest MIC values at 0.01 µg/mL for both. The BP-antibiotic conjugate with the lowest and most favorable MIC profile was BCN, followed by BCS, with the etidronate conjugates demonstrating the highest MIC profiles. The parent antibiotics exhibited slightly higher effectiveness than the conjugates as expected in these experimental settings, since the structural modifications of the conjugates usually reduce the activity and the in vitro assay conditions used here may not provide the required environment for the sufficient cleavage of the linker and release of the parent antibiotics. The results obtained from bio-screen assays confirmed our real-time data and further demonstrated that the antibacterial properties of the conjugates come from the released antibiotic and not from the BP moiety of the conjugates.

The addition of HA to the parent antibiotics and the BP-antibiotic conjugates did not change the patterns of inhibition. In fact, the addition of HA resulted in higher MIC values for both the parent antibiotics and the conjugates, an observation that was also confirmed by our bio-screen results. The presence of HA might reduce the contact and the exposure of the testing compound to the biofilm resident bacteria in this type of kinetic experimental apparatus and setting and the cleavage of these linkers to release active antibiotics may not occur rapidly. HA in general does not have appreciable antibacterial properties, which we confirmed in our data [22]. HA may also influence the impedance values due to the presence of numerous charged ions in the solution; HA has not been tested before in such an experimental setting with impedance assays. BPs themselves do not interfere with the antibacterial effect of the conjugates, as we have also previously demonstrated [12]. After evaluating the MIC values of parent antibiotics and the BP-antibiotic conjugates, the bone-binding property of the conjugates was tested with a concentration-dependent method, and it was established that the BP-antibiotic conjugates did have the ability to bind to HA. Therefore, despite higher MIC values of the conjugates versus parent antibiotics, the ability to target bone for the BP-antibiotic conjugates as compared to parent drugs should allow for higher concentrations of the antibiotic at infected bone sites with less systemic exposure, which is important for clinical translation and minimizing systemic toxicity. The research into bone-seeking medicinal agents such as the work presented here is progressively laying the foundation for next-generation BP “target and release” type ‘magic bullets’ that minimize systemic exposure and maximize drug efficacy at the targeted site.

## 4. Materials and Methods

### 4.1. Synthesis

In this study, quinolone antibiotics sitafloxacin, nemonoxacin, ciprofloxacin, and moxifloxacin were conjugated with either [2-(4-hydroxyphenyl)ethane-1,1-diyl]bis(phosphonic acid) (HPBP) or etidronic acid via a releasable carbamate linker. The structures of the compounds used in this study are shown below.

For the synthesis of the HPBP–quinolone conjugate BCS (BV600072) and BCN (BV600082), we used a modified version of the scheme that we previously used to synthesize the ciprofloxacin conjugate BCC [12]. The synthesis of these conjugates is shown in Figure 1. Briefly, tetraethyl methylenebis(phosphonate) was condensed with 1-(Benzyloxy)-4-(bromomethyl)benzene. The benzyl group was removed using catalytic hydrogenation and the tetraethyl-protected BP fragment was converted to p-nitrophenol active ester using p-nitrophenyl chloroformate. Next, the ethyl esters were removed using BTMS and the deprotected p-nitrophenyl BP fragment was reacted with sitafloxacin and nemonoxacin in the presence of Hunig’s base to provide BCS and BCN, respectively (Figure 1).

The synthesis of etidronic acid conjugates ECC and ECX was enabled by the preparation of tetramethyl-protected etidronate [17]. The synthesis of these conjugates is described in Figure 2. Briefly, acetyl chloride was mixed with trimethyl phosphite to provide dimethyl acetylphosphate that was further reacted with dimethyl phosphite to provide tetramethyl etidronate. The tetramethyl etidronate was converted to the p-nitrophenyl active ester using p-nitrophenyl chloroformate. The reaction of the p-nitrophenyl active ester with fluoroquinolones ciprofloxacin and moxifloxacin provided the tetramethyl-protected conjugates. The final compounds ECC and ECX were obtained by the removal of the methyl groups using BTMS. The syntheses and characterization data of these conjugates are described in Figure 2. 

### 4.2. Experimental Strain and Culture Conditions

For experimental purposes, *Staphylococcus aureus* (*S. aureus*) ATCC 6538 was used. As a standard culture condition, *S. aureus* was cultured in trypticase soy broth with 0.6% yeast extract (TSBYE).

### 4.3. Antimicrobials Tested

The following antibiotics were tested: ciprofloxacin (C), moxifloxacin (X), sitafloxacin (S), and nemonoxacin (N). The following BP-antibiotic conjugates were tested: bisphosphonate-carbamate-sitafloxacin (BCS, BV600072), bisphosphonate-carbamate-nemonoxacin (BCN, BV600082), etidronate-carbamate-ciprofloxacin (ECC, BV81022), and etidronate-carbamate-moxifloxacin (ECX, BV81051).

### 4.4. Monitoring of Biofilm Formation in Real Time

The formation of biofilm was monitored using a real-time cell analyzer (RTCA) xCELLigence (ACEA Bioscience Inc., SanDiego, CA, USA) instrument [18]. Next, 80 µL of TSBYE was added to each well of non-reusable 16X microtiter E-plates (ACEA Biosciences) for the impedance background measurement using the standard protocol provided by the software. Then, standardized overnight grown culture was diluted to 10^7^ CFU/mL in fresh TSBYE broth and was then added to 16 E-plate wells in a total of 120 µL of TSBYE. Each sample was run in duplicate. The E-plates were positioned in the xCELLigence Real-Time Cell Analyzer MP, incubated at 37 °C, and monitored on the RTCA system at 15 min time intervals for 24 h. Another real-time analysis was performed using a bioscreen plate reader to confirm the results from the xCELLigence RTCA. For this purpose, the overnight grown biofilm was added to the plate and diluted with TSBYE to a total volume of 200 µL and then shaken for 24 h with readings every 15 min.

### 4.5. Optimizing the Cell Suspension Concentration

To determine the most optimal culture configuration that provides the most precise graph, 80 µL of TSBYE was added to each well for the impedance background measurement. The biofilm-forming strain, which was grown overnight and diluted down to 10^7^ CFU/mL in fresh TSBYE broth, was added to the 16 E-plate wells, in a total of 120 µL of TSBYE with further dilutions of bacteria as follows: undiluted, 1:5, 1:10, 1:25, 1:50, 1:100, and 1:200.

### 4.6. Monitoring the Biofilm Inhibition Effect of BP-Antibiotic Conjugates and the Parent Antibiotics

The xCELLigence RTCA MP instrument (ACEA Biosciences) was utilized for all impedance experiments. First, 80 uL of cell culturing media was added to each well of the 96 well E-Plates (Agilent Technologies; Santa Clara, CA, USA) and the background impedance was measured and displayed as the Cell Index. The Cell Index (CI) is a measure of the relative change in the electrical impedance from the initial background reading at a certain frequency (ƒn). After the addition of 80-µL of TSBYE to each well to measure the background impedance, 1 µL of bacterial suspension in a total of 120 µL of TSBYE containing a range of concentrations of antibiotic was added to the 16 E-plate wells. Two replicates of each antibiotic concentration and negative controls without antibiotics were also included. The cells were monitored for 24 h and then the analysis was performed.

The Cell Index at a given time point t CI(t) is calculated as follows:CI(t) = [R(ƒ, t) − R(ƒ, t0)]/Z
where:

ƒ is the frequency that the impedance measurement is carried out at (10 kHz).

R(ƒ, t) is the measured impedance at frequency ƒ at time point t.

R(ƒ, t0) is the measured impedance at frequency ƒ at time point t0 (usually t0 is the time when the background is measured).

Z is the corresponding frequency factor, which for 10 kHz is 15.

Thus, the cell index is directly related to the change in the impedance from the initial background reading, which is set to 0.

The Delta Cell Index function adds a constant value to the Cell Index from each well, so they have the same Delta Value at the reference time point.

In simple terms, the Delta Cell Index shifts each well Cell Index, so they have the prefixed value at the selected Delta Time.

To calculate the Delta Cell Index, the Cell Index values of the selected wells at a reference time point (i.e., Delta Time or tDelta) are set to a constant (Delta Constant). The differences between this chosen constant and the original Cell Index is called the Delta Value and is calculated as follows:

DeltaValuewell_i = Delta Constant − CIwell_i(tDelta) with i = 1, 2, …, n. At a given time point t, the Delta Cell Index (DCI(t)) is calculated as the sum of the Cell Index at time point t and the Delta Value as follows:

DCIwell_i(t) = CIwell_i(t) + DeltaValuewell_i = CIwell_i(t) − CIwell_i(tDelta) + Delta Constant where, DCIwell_i(t) is the Delta Cell Index of well i at time point t. CIwell_i(t) is the Cell Index of well i at time point t. CIwell_i(tDelta) is the Cell Index of well i at Delta Time tDelta.

### 4.7. Minimum Inhibitory Concentration Measurement

In order to compare the efficacy of different BP-antibiotic conjugates and the parent antibiotics, a numerical parameter was needed. The minimum inhibitory concentration (MIC) of four BP-antibiotic conjugates and the parent antibiotics for the tested strain was calculated through data analysis via the xCELLigence RTCA system.

### 4.8. Affinity of BP-Antibiotic Conjugates to Hydroxyapatite

In order to study the HA binding affinity of the BP-antibiotic conjugates and parent antibiotics, we performed HPLC analysis as follows: 500 µg/mL of each testing compound (all the BP-antibiotic conjugates and parent compounds) was added to a solution containing 5 mg/mL of HA spherules and incubated at 37 °C/4 h under magnetic stirring. Next, the mixture was allowed to sediment for 1 h/4 °C. After that, the quantity of remaining drug in the supernatant was assessed using HPLC (AGILENT 1220 Infinity LC system; Santa Clara, CA, USA); the gradient was applied: phase A—20% CH3CN in 0.1 M NH_4_OAc, pH = 7 and phase B: 70% CH_3_CN in 0.1 M NH_4_OAc, pH = 7. The column was applied: Supelco ascentris expressed C18: 15 cm × 4.6 mm × 5 µm. For ECC and C, it was isochoric: 0–20 min—100% of phase A. For ECX and X, the gradient was: 0–3 min—0–10% phase B, 0–3 min—0–10% phase B, 3–5 min—10% phase B, 5–10 min—10–30% of phase B, and 10–15 min—30% of phase B. For BCS, BCN, S, and N, the gradient was: 0–7 min—0% phase B, 7–25 min—0–100% phase B, and 25–40 min—100% phase B. To evaluate the percentage of BP-antibiotic conjugate bound to HA, we used the methodology described in detail in a previous publication [12]. Additionally, 500 µg/mL of each compound without incubation with HA served as a control sample for testing. The affinity of compounds to HA was estimated as follows: 100%—peak area of tested sample in supernatant/ peak area of control sample in supernatant * 100%.

### 4.9. Infection Preventative Experiment in the Presence of HA via Real-Time Analysis

Different concentrations of each test compound were added to a solution containing 10 µg/mL of HA powder and incubated for 2 h/37 °C under magnetic stirring. Then, 80 µL of TSBYE was added to each well to measure the background impedance. Next, 1 µL of bacterial suspension in a total of 120 µL of TSBYE containing a gradient of concentrations of testing compounds plus HA was then added to the 16 E-plate wells. Two replicates of each testing compound concentration and negative controls without a testing compound were also included. The cells were monitored for 24 h and analysis was performed to calculate the MIC50 values.

### 4.10. Growth Analysis Using Bio-Screen C Automated Microbiology Growth Curve Analysis System

To confirm the growth trends we observed from xCELLigence RTCA, another real-time analysis was performed using a Bio-Screen C automated microbiology growth curve analysis system (Growth Curve USA, Piscataway, NJ, USA). For this analysis, the overnight biofilm growth was diluted with TSBYE to a total volume of 200 µL and loaded into each well of a 100-well honeycomb microplate. The turbidity of the culture in each well was then measured continuously using the 600 nm filter with continuous shaking and 24- or 48 h growth curves were then obtained and compared to the xCELLigence RTCA data.

### 4.11. Preventative Antimicrobial Assays through Spectroscopic Analysis

A variety of concentrations of BP-antibiotic conjugates were introduced to the HA spherules suspended in TSBYE medium and incubated for 24 h/37 °C under magnetic stirring. Next, the supernatant was removed gently and fresh *S. aureus* with a density of 105 CFU/mL was added to the HA spherules. This mixture was shaken and the absorbance was measured at 580 nm wavelength and left for incubation for 24 h/37 °C/shaking. The absorbance was then measured again after incubation. The two control samples were also used wherein control sample one was a bacterial suspension without HA and control sample two was a bacterial suspension + HA spherules with no BP-antibiotic conjugate added.

### 4.12. Statistical Analyses

The statistical analyses were performed using GraphPad Prism 8.0.1 and 9.3.1 (GraphPad Software, San Diego, CA, USA). The normality of distribution was verified using a Shapiro–Wilk’s test. To evaluate the statistical significance, an ANOVA test with a post hoc Tukey modification (α = 0.05) was performed.

## 5. Conclusions

Real-time biofilm analysis allows us to detect changes in microbial mass over time during antimicrobial therapy, which can be a promising tool to clinically evaluate antibiotic susceptibility and efficacy in biofilm-mediated infections. The novel BP-antibiotic conjugates designed and tested in this study retained the bone binding properties of the BP moiety. They also retained the antimicrobial activity of the parent antibiotic in the presence or absence of HA, albeit at lower levels due to the nature of their chemical modification. These results demonstrate the utility of real-time impedance assays in the drug screen toward biofilm inhibition, confirm previous findings in this field, and provide an additional dataset that will aid us in chemistry optimization for antimicrobial effects in future iterations of the conjugates in this class as we prepare for clinical testing. This aims to achieve the optimized drug concentration, bone binding affinity, and release kinetics. This class of BP-antibiotic conjugates using BPs as biochemical vectors for the delivery of antibiotic agents to bone (where osteomyelitis biofilm pathogens reside) could represent an advantageous approach to the treatment of osteomyelitis attributing to their improved bone pharmacokinetics and minimized systemic exposure or toxicity to host or eukaryotic cells.

## Data Availability

The authors confirm the data supporting the findings of this study are available within the article. Raw data that support the findings of this study are available from the corresponding author upon reasonable request.

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
