# Peer review of "Real-Time Impedance-Based Monitoring of the Growth and Inhibition of Osteomyelitis Biofilm Pathogen Staphylococcus aureus Treated with Novel Bisphosphonate-Fluoroquinolone Antimicrobial Conjugates"

_ijms, 2023, doi:10.3390/ijms24031985_

Round 1

Reviewer 1 Report

The manuscript entitled" Real-time impedance-based monitoring of the growth and inhibition of osteomyelitis biofilm pathogen Staphylococcus aureus treated with novel bisphosphonate-fluoroquinolone antimicrobial conjugates" uses an attractive methodology (impedance-based) to investigate an important issue which is the design of biochemical vectors for the delivery of antibiotic agents to bone minerals. While I do not have any concerns about the scientific part of the described research, I do have some issues about the methodology and the way of presenting results.

As an electrochemist, I admit that impedance is a complex property. Impedance investigations are extremely useful in the variety of fields, but they are also not easy to plan and interpret. The main problem with any impedance-based analysis method is the fact, that they are extremely easy to perform, but only makes sense if are performed well. It is true that impedance can be used to non-invasive monitoring of biofilm status including cell number or index, shape and size, and attachment – all without the need for labeling. But, to be honest, an extensive processing of resulting data is required to be able to properly analyzed aforementioned characteristics.

1. The Authors are advised to add to the introduction section (after lines 95-101) a paragraph summarizing the challenges and limitations associated with impedance measurements, including defined experimental conditions, state of cells, and the fact that the impedance provides an overall information about the state of biofilm, and it is very hard to say what really happened to the system when the change i impedance is observed.

2. The methodology section should be extended with more detailed description of impedance measurements, particularly delta cell index should be explained - how is it calculated? What is its relation to impedance?

3. Figures should be improved: the time axis in fig 2 and fig 3 should be corrected to include time integers. Putting the legend on figures (e.g. abbreviations and concentrations) would make it easier for the readers to  follow the graphs.

4. Figure numbers in the text should be double-checked.

5. Figure 5 have wrong color labelling - different colours in the figure, and in the legend. Significance should be tested and marked in the graph.

6. Discussion section should also cover the disadvantages and limitations of impedance-based measurements (related to his study).

Author Response

Reviewer: 1

Comments to the Author
The manuscript entitled" Real-time impedance-based monitoring of the growth and inhibition of osteomyelitis biofilm pathogen Staphylococcus aureus treated with novel bisphosphonate-fluoroquinolone antimicrobial conjugates" uses an attractive methodology (impedance-based) to investigate an important issue which is the design of biochemical vectors for the delivery of antibiotic agents to bone minerals. While I do not have any concerns about the scientific part of the described research, I do have some issues about the methodology and the way of presenting results.

As an electrochemist, I admit that impedance is a complex property. Impedance investigations are extremely useful in the variety of fields, but they are also not easy to plan and interpret. The main problem with any impedance-based analysis method is the fact, that they are extremely easy to perform, but only makes sense if are performed well. It is true that impedance can be used to non-invasive monitoring of biofilm status including cell number or index, shape and size, and attachment – all without the need for labeling. But to be honest, an extensive processing of resulting data is required to be able to properly analyze aforementioned characteristics.

Thank you for these valuable comments we agree, and we hope you will find the revised manuscript more readable since we further discuss the nature of such data and interpretations.

  1. The Authors are advised to add to the introduction section (after lines 95-101) a paragraph summarizing the challenges and limitations associated with impedance measurements, including defined experimental conditions, state of cells, and the fact that the impedance provides an overall information about the state of biofilm, and it is very hard to say what really happened to the system when the change in impedance is observed.
    Excellent comment and revised as requested by adding such wording to the introduction.

  2. The methodology section should be extended with more detailed description of impedance measurements, particularly delta cell index should be explained - how is it calculated? What is its relation to impedance?
    As requested, this has been revised with an additional paragraph in the methodology section detailing the impedance measurements and values.

  3. Figures should be improved: the time axis in fig 2 and fig 3 should be corrected to include time integers. Putting the legend on figures (e.g. abbreviations and concentrations) would make it easier for the readers to follow the graphs.
    Revised as requested thank you.

  4. Figure numbers in the text should be double-checked.
    Revised as requested.

  5. Figure 5 have wrong color labelling - different colours in the figure, and in the legend. Significance should be tested and marked in the graph.
    Revised as requested.

  6.  Discussion section should also cover the disadvantages and limitations of impedance-based measurements (related to his study).
    Great comment and revised as requested by adding another paragraph to the discussion section covering the limitations and disadvantages of impedance measurements.

Thank you for your consideration!

Reviewer 2 Report

The study by Sedghizadeh et al. is focused on the development of antimicrobial conjugates effective against Staphylococcus aureus. Although the developed system seems to affect the growth of the pathogens thus preserving the antibiotic activity of the conjugated molecules, overall the experimental data do not show any proof of “anti-biofilm” activity. The authors should clarify that all the collected results are indicative of activity against planktonic bacterial cells (detectable by Absorbance at 600 nm) and not against bacterial biofilm, which needs different experimental conditions (e.g. inoculum higher than 107 CFU/mL; staining procedures with for example crystal violet or SYTO-9; antibiotic concentration lower than MIC100) in order to be detected. Also, the data representation is unclear (for example, the legends should be included in the panel and not in the figure description; the acronymous of each test compound should be introduced in the main text before the figure; each graph should have a title). The authors should define the arbitrary units used (e.g. Delta cells index) and add standard deviation and statistics. Moreover, words such as S. aureus, in vitro, and in vivo should be written in italics. The manuscript is not ready to be published in its current form.

Author Response

Reviewer: 2

Comments to the Author

 The study by Sedghizadeh et al. is focused on the development of antimicrobial conjugates effective against Staphylococcus aureus. Although the developed system seems to affect the growth of the pathogens thus preserving the antibiotic activity of the conjugated molecules, overall the experimental data do not show any proof of “anti-biofilm” activity. The authors should clarify that all the collected results are indicative of activity against planktonic bacterial cells (detectable by Absorbance at 600 nm) and not against bacterial biofilm, which needs different experimental conditions (e.g. inoculum higher than 107 CFU/mL; staining procedures with for example crystal violet or SYTO-9; antibiotic concentration lower than MIC100) in order to be detected.

Thank you for the great comments, and we have added similar wording to the manuscript to indicate planktonic versus biofilm results.

Also, the data representation is unclear (for example, the legends should be included in the panel and not in the figure description; the acronymous of each test compound should be introduced in the main text before the figure; each graph should have a title).

Thank you for these valuable comments to improve the data and figure presentations, and we have revised accordingly as recommended.

The authors should define the arbitrary units used (e.g. Delta cells index) and add standard deviation and statistics.

Thank you great comment and we have edited as requested by defining the cell indices in the revised manuscript.

Moreover, words such as S. aureus, in vitro, and in vivo should be written in italics.

Revised as requested.

Thank you for your consideration!

Round 2

Reviewer 1 Report

I appreciate the revision provided by Authors, particularly a detailed definition of cell index and delta cell index.

Author Response

Reviewer: 1

Comments to the Author
I appreciate the revision provided by Authors, particularly a detailed definition of cell index and delta cell index.

Thank you so much for your time and all valuable comments that improved the manuscript and made it more understandable.

Reviewer 2 Report

The authors have addressed the comments provided during the first revision. However, the following remaining edits should be accomplished before the manuscript consideration:

·       Figure 2: S. aureus should be written in italic within the legend.

·       Figure 1-2-3-4: the authors should add standard deviations.

·       Caption of figure 4: the author should clarify that the detected signal (OD) is related to bacterial growth and not the biofilm.

·       Line 430: the author should add a reference at the end of the sentence “the traditional spectroscopic analysis”.

·       Line 433: the author should replace the term “biofilm” with “bacterial”.

·       Line 441-443 and Figure 5: the authors should add proof of the statement “…because bacteria bind to HA and form biofilms which are not measured in the HA-free supernatant” or delete the sentence. Moreover, the antimicrobial efficacy of each treated sample should be compared to both “Ct” and “Ct + HA” samples. The authors should define statistical differences between treated groups and each control.

·       Since the authors have added descriptions, and extra details in the methodology about the function of the xCELLigence RTCA MP instrument (ACEA Biosciences) in monitoring biofilm inhibition, they should edit the text as follows

o   Line 471: “bacterial growth and biofilm inhibition” instead of “bacterial formation and inhibition”

o   Line 475: “bacterial growth” instead of “bacterial colony formation”

o   Line 476: “biofilm formation” instead of “bacterial formation”

o   Line 480: “Bacterial growth and biofilm formation” instead of “bacteria or biofilm formation”

Author Response

Reviewer: 2

Comments to the Author

The authors have addressed the comments provided during the first revision. However, the following remaining edits should be accomplished before the manuscript consideration: Thank you for your positive comments and we have edited as recommended.

 Figure 2: S. aureus should be written in italic within the legend. Revised as requested.

Figure 1-2-3-4: the authors should add standard deviations. Revised as requested.

Caption of figure 4: the author should clarify that the detected signal (OD) is related to bacterial growth and not the biofilm. Revised as requested.

Line 430: the author should add a reference at the end of the sentence “the traditional spectroscopic analysis”. Revised as requested.

Line 433: the author should replace the term “biofilm” with “bacterial”. Revised as requested.

Line 441-443 and Figure 5: the authors should add proof of the statement “…because bacteria bind to HA and form biofilms which are not measured in the HA-free supernatant” or delete the sentence. Moreover, the antimicrobial efficacy of each treated sample should be compared to both “Ct” and “Ct + HA” samples. The authors should define statistical differences between treated groups and each control. Revised as requested.

Since the authors have added descriptions, and extra details in the methodology about the function of the xCELLigence RTCA MP instrument (ACEA Biosciences) in monitoring biofilm inhibition, they should edit the text as follows

Line 471: “bacterial growth and biofilm inhibition” instead of “bacterial formation and inhibition” Revised as requested.

Line 475: “bacterial growth” instead of “bacterial colony formation” Revised as requested.

Line 476: “biofilm formation” instead of “bacterial formation” Revised as requested.

Line 480: “Bacterial growth and biofilm formation” instead of “bacteria or biofilm formation” Revised as requested.

Thank you for your consideration!

Round 3

Reviewer 2 Report

The revised version of the manuscript can be accepted in its present form.